# Phenology and Dwarfing Gene Interaction Effects on the Adaptation of Selected Wheat (*Triticum aestivum* L.) Advanced Lines across Diverse Water-Limited Environments of Western Australia

**Mirza A.N.N.U. Dowla** [1], **Shahidul Islam** [1], **Katia Stefanova** [2], **Graham O' Hara** [1], **Wujun Ma** [1] **and Ian Edwards** [1,3,*]

[1] College of Science Health Engineering and Education, Murdoch University, 90 South Street, Murdoch, WA, Perth 6150, Australia; Mirza.Dowla@murdoch.edu.au (M.A.N.N.U.D.); s.islam@murdoch.edu.au (S.I.); G.Ohara@murdoch.edu.au (G.O.H.); W.Ma@murdoch.edu.au (W.M.)

[2] The UWA Institute of Agriculture, Faculty of Science, University of Western Australia, 35 Stirling Hwy, Crawley, WA, Perth 6009, Australia; katia.stefanova@curtin.edu.au

[3] Edstar Genetics Pty. Ltd., Murdoch University, 90 South Street, Murdoch, WA, Perth 6150, Australia

[*] Correspondence: ian@edstargenetics.com; Tel.: +614-08-9387-4537 or +614-08-93606323

**Abstract:** Photoperiod, vernalization, and plant height controlling genes are major developmental genes in wheat that govern environmental adaptation and hence, knowledge on the interaction effects among different alleles of these genes is crucial in breeding cultivars for target environments. The interaction effects among these genes were studied in nineteen Australian advanced lines from diverse germplasm pools and four commercial checks. Diagnostic markers for the *Vrn-A1* locus revealed the presence of the spring allele *Vrn-A1a* in 10 lines and *Vrn-A1c* in one line. The dominant alleles of *Vrn-B1a* and *Vrn-D1a* were identified in 19 and 8 lines, respectively. The most common photoperiod-insensitive allele of *Ppd-D1a* was identified in 19 lines and three and four copy photoperiod-insensitive alleles (*Ppd-B1a* and *Ppd-B1c*) were present in five and one lines, respectively. All the lines were photoperiod-sensitive for the *Ppd-A1* locus. All lines were semi-dwarf, having either of the two dwarfing alleles; 14 lines had the *Rht-B1b (Rht-1)* and the remaining had the *Rht-D1b (Rht-2)* dwarfing allele. The presence of the photoperiod-insensitive allele *Ppd-D1a* along with one or two spring alleles at the *Vrn1* loci resulted in an earlier heading and better yield. Dwarfing genes were found to modify the heading time—the *Rht-D1b* allele advanced heading by three days and also showed superior effects on yield-contributing traits, indicating its beneficial role in yield under rain-fed conditions along with an appropriate combination of photoperiod and vernalization alleles. This study also identified the adaptability value of these allelic combinations for higher grain yield and protein content across the different the water-limited environments.

**Keywords:** vernalization; photoperiod; dwarfing gene; water-limited environments; adaptation; stability

---

## 1. Introduction

Australia is the fourth largest wheat exporter of the world, with 40–50% of its production coming from Western Australia [1]. As in many other countries, wheat is grown under rain-fed conditions in Australia [2]. Therefore, yield potential depends mainly on the environmental conditions during the growing season, which include temperature and rainfall, as well as heat and frost events. Generally, in Western Australia, wheat is sown after the first flush of rain in late autumn or early winter, seeking to

sow late enough to escape frost damage during flowering in spring but early enough to allow plants to reach physiological maturity before the beginning of the dry hot summer [3]. Control of phenological development is governed mainly by vernalization and photoperiod responsive genes, which play key roles for the successful adaption to the target environments. Southwestern Australia is characterised by a Mediterranean climate, classified as semi-arid dryland [4]. Late maturing tall wheat varieties were formally confined to wetter long growing season areas. These varieties have been replaced by photoperiod and gibberellin-insensitive varieties better suited to Australian conditions and which have allowed wheat to be grown in drier environments [3,5].

Due to global warming, the climate of southwestern Australia is expected to become warmer with the increase of temperature by 1.25–1.5 and 1.5–1.75 °C in the coastal regions, the west, and the more inland east [2]. The cumulative effects of changing temperature and rainfall will increase the frequency of drought episodes and affect wheat production in low latitudes, including Australia, more than in high latitudes. A 2 °C increase in temperature could depress wheat yields by up to 40% [6] unless we adopt appropriate, improved varieties and good management practices in a timely fashion and as a matter of urgency. Improved varieties will require fine-tuning of phenological development in order to adapt wheat production to future changing climate conditions in southwest Australia.

Flowering in wheat is primarily controlled by at least five vernalization loci (*Vrn-A1*, *Vrn-B1*, *Vrn-D1*, *Vrn-2*, and *Vrn-3*), three homoeologous loci of photoperiod genes (*Ppd-D1*, *Ppd-B1*, and *Ppd-A1*), and earliness *per se* genes [7,8]. With recent advances in molecular biology, a number of alleles at the *Ppd* and *Vrn1* loci, including haplotypes and copy number variations, have been identified as being responsible for affecting heading date by modifying phenological development phases and other agronomic traits [9–20]. This allelic variation is associated with insertions, deletions, and mutations in the promoter region of *Vrn1*, and deletion or transposon insertion within the promoter region and also copy number variation for the *Ppd* gene [9,11,20]. All these alleles of vernalization and photoperiod genes respond differently to environmental stimuli and act initially within separate pathways, which converge at a point to produce flowers [21,22]. Thus, each of these alleles has adaptive value to specific environments, whereby 70–75%, 20–25%, and 5% genetic variability have been attributed to vernalization, photoperiod, and earliness *per se* genes, respectively [23]. On the other hand, the dwarfing genes (*Rht*), acknowledged as the genetic basis of the green revolution during the 1970s, are also known to interact with the phenology genes in determining yield [24]. The availability of molecular markers for those alleles makes it easy to identify and trace them in breeding populations [25,26]. Quantification of the effects of different allelic combinations of *Vrn*, *Ppd*, and *Rht* genes on heading date could provide a guideline for the strategic breeding of wheat varieties for specific water-limited environments via drought stress avoidance.

The overall objective of this study was to identify and estimate the interaction effects of *Vrn1* and *Ppd* allelic combinations on heading date and yield parameters in selected advanced breeding lines developed from diverse parents among five different germplasm pools. In the analysis, all the three homoeologous loci of *Vrn1* and *Ppd* along with *Rht-1* and *Rht-2* genes were included to obtain accurate estimates of the genetic and environmental interaction effects of different alleles on heading date and other agronomic traits, with the goal of providing useful data for wheat breeding programs targeting specific water-limited environments. A further objective was to determine the most favourable allelic combinations for different wheat growing regions of Western Australia.

## 2. Materials and Methods

### 2.1. Plant Materials

Nineteen advanced lines with various allele combinations in five different genetic pools and four commercial checks were used for this study (Table 1). The genetic pools represented diverse genotypes from Spanish x French (Europe), Queensland, Synthetics (Victoria), CIMMYT (Mexico), and Winter X Spring crosses using UK winter varieties, as presented in Table 2. Despite the wide genetic diversity of the parents, the breeding lines went through routine screening for yield, quality, disease resistance, and overall agronomic performance against commercial check cultivars and proved to be well adapted in several years of preliminary trials. Four local check cultivars were Wyalkatchem, Magenta (Australian Premium white classification), Mace, and Bonnie Rock (Australian Hard wheat classification).

Mace has a Wyalkatchem genetic background with higher grain yield, which has led to a rapid uptake across the environments of WA. Mace also provides good disease resistance, grain quality, and better tolerance to sprouting compared to Wyalkatchem and Magenta.

Wyalkatchem has been the most widely adapted variety and good yielder in the water-limited regions of WA and has a good level of tolerance to acid soils. Wyalkatchem is resistant to yellow spot (*Pyrenophora tritici-repentis*), hence being suitable for wheat-on-wheat systems. Magenta is a mid-long maturing variety and is best suited to early sowing. Its yield is similar to Wyalkatchem; it is also resistant to yellow spot and has good early vigour due to a longer coleoptile. Bonnie Rock is an early maturing variety and is known for its good baking quality and has also a good resistance to stem rust (*Puccinia graminis*).

### 2.2. Field Experiments

Field experiments were conducted in 2014 and comprised the 23 lines, which were grown under rainfed conditions in three locations across WA, namely Kojonup (32.7° S–117.4° E), Corrigin (32.3° S–117.8° E), and Toodyay (31.4° S–116.5° E). The lines were sown using randomised complete block design with three replications in May following the first flush of rain in 6.0 × 1.35-m plots. Daily meteorological data were obtained from the nearest Bureau of Meteorology (BOM) station (Table S1). Nutrient and pest management practices were done according to local farmers' standard practice.

### 2.3. Agronomic Traits

In the field trials, grain harvested (Table 3) from each plot was converted to tons/ha yield. Data for heading, plant height, and physiological maturity were recorded for the Toodyay site. Heading time was recorded when 50% of spikes had emerged from the flag leaf, and anthesis was determined when 50% of the spikes had extruded anthers. Physiological maturity was recorded when 50% of the culm below the spikes had turned yellowish. Plant height was measured from the soil surface to the top of the spike without including the awn.

Twenty main heads were harvested from each plot and spikes were measured and threshed manually to obtain the data for spike length and grains per spike. Then, data for thousand-seed weight, test weight (hectolitre), seed length, width, plumpness, and roundness were taken using a digital seed image analyser (SeedCount™ version 2.4.0, Next Instruments, Australia) in the seed testing laboratory of the Department of Agriculture and Food (DAFWA), South Perth, WA.

## 2.4. Genotyping of the Plant Materials

Genomic DNA was extracted from leaf tissues of 10-day old seedlings of each line, including controls using SDS extraction protocol. *Vrn1* gene alleles were identified using the primers described by Yan et al. [20] and Fu et al. [26]. The spring allele *Vrn-A1a* was identified using the primer combination VRN1AF and VRN-INT1R. All the lines were tested using three pairs of primers to distinguish between the presence of the dominant allele *Vrn-A1c* and recessive allele *Vrn-A1v*. The dominant *Vrn-B1a* allele was identified using the primer pair Intr1/B/F and Intr1/B/R3. The primer pair Intr1/D/F and Intr1/D/R3 were used to identify the presence of the dominant *Vrn-D1a* allele and recessive *Vrn-D1v* was identified using the primer pair Intr1/D/F and Intr1/D/R4. *Ppd* alleles were identified using the primers developed by [9]. Multiplex PCR with primers Ppd-A1proF/durum_Ag5del_F2/durum_Ag5del_ R2 [14] generated a 452 bp fragment characteristic of the recessive *Ppd-A1b* allele (Table 1). For the identification of the *Ppd*-B1 allele, the lines were tested against two sets of primers according to Díaz et al. [11]. Alleles of *Ppd-D1* were identified using multiplex PCR with primers Ppd-D1_F/Ppd-D1_R1/Ppd-D1_R2 [9]. Two sets of primers were used to identify the alleles for reduced height at the *Rht-B1* and *Rht-D1* loci using primers and protocol followed by Ellis et al. [27]. In brief, PCR was performed for *Vrn* and *Ppd* genes following 2 min denaturation at 94 °C, samples were subjected 35 cycles in a Touch down (TD) program (94 °C for 30 s, 60 °C for 30 s, and 72 °C for 1 min, followed by a 1 °C decrease in annealing temperature in every cycle for first 5 cycles), ending with a 7 min extension at 72 °C. A similar TD PCR program was conducted for Rht genes, with the exception of starting annealing temperature at 63 °C. PCR products were visualised in 1% agarose gel and specific alleles were identified based on respective band size. Designation of the vernalization, photoperiod, and reduced height alleles was adopted from Eagles et al. [25], Fu et al. [26], Ellis et al. [27], and Cane et al. [28]. For ease of discussion, allelic combinations have been represented by seven letters, where the first three letters designate spring (S) or winter (W) alleles at the *Vrn-A1*, *Vrn-B1*, and *Vrn-D1* loci, respectively, the next two letters designate the photoperiod-insensitive alleles (A) or -sensitive alleles (B) at the *Ppd-D1* and *Ppd-B1* loci, respectively, and the last two letters designate dwarf (D) or tall (T) alleles at the *Rht-B* and *Rht-D* loci. Since all lines were recessive for the *Ppd-A1* locus, this information was not included in the analysis (Table 1).

## 2.5. Statistical Analysis

A more complex linear mixed model was adopted in the current research, where the GxE effect for yield was modelled using a Multiplicative Mixed Model (MMM); more specifically, this is a Factor Analytic (FA) model [29], accounting for GxE and for heterogeneous genetic variance and covariance between trials.

In the current study, the data for yield and protein did not have a complete spatial configuration; therefore, a general LMM model [29] was used to model GxE interactions. The latter involves a variance component model fitting environment and variety/AC main effects and varietal/AC interactions with the environment (trial), referred to as GxE.

All single-site and MET analyses involved model selection based on the Log Likelihood and Akaike Information Criterion (AIC).

The dataset was analysed using GenStat 20 and ASREML-R [30], which facilitates joint modelling of blocking structure, spatial variation, treatment effects, and extraneous variation. The Additive Main Effects and Multiplicative interactions (AMMI) stability value (ASV) is a measure of the distance of a genotype from the origin in a two-dimensional scatter diagram of IPCA1 scores against IPCA2 scores, as proposed by Purchase et al. [31]. A smaller ASV value indicates a more stable genotype over different environments. On the other hand, according to Eberhart and Russell [32], genotypes with regression coefficient one ($b_i = 1$) and squared deviation from regression zero ($S^2di = 0$) are more stable and widely adaptable. The AMMI model and stability analysis was performed using the R package plantbreeding (V1.1.1) [33].

**Table 1.** Allelic composition of the advanced lines.

| SL. NO. | ALLELIC COMBINATION [i] | Line Name | Vernalization Loci *Suffix* a and c = "S" and v = "W" | | | Photoperiod Loci *Suffix* a and c = "A" and b = "B" | | | Reduced Height Loci *Suffix* a = "T" and b = "D" | |
|---|---|---|---|---|---|---|---|---|---|---|
| | | | *VRN A1* | *VRN B1* | *VRN D1* | *PPD-A1* | *PPD-D1* | *PPD-B1* | *Rht-1* | *Rht-2* |
| 1 | SSS-AB-DT | CMT-4 | *Vrn-A1a* | *Vrn-B1a* | *Vrn-D1a* | *Ppd-A1b* | *Ppd-D1a* | *Ppd-B1b* | *Rht-B1b* | *Rht-D1a* |
| 2 | SSW-AB-DT | QLD-4<br>Bonnie Rock | *Vrn-A1a*<br>*Vrn-A1a* | *Vrn-B1a*<br>*Vrn-B1a* | *Vrn-D1v*<br>*Vrn-D1v* | *Ppd-A1b*<br>*Ppd-A1b* | *Ppd-D1a*<br>*Ppd-D1a* | *Ppd-B1b*<br>*Ppd-B1b* | *Rht-B1b*<br>*Rht-B1b* | *Rht-D1a*<br>*Rht-D1a* |
| 3 | SSW-AB-TD | SP-2<br>VIC-1<br>VIC-2<br>VIC-3 | *Vrn-A1a*<br>*Vrn-A1a*<br>*Vrn-A1a*<br>*Vrn-A1c* | *Vrn-B1a*<br>*Vrn-B1a*<br>*Vrn-B1a*<br>*Vrn-B1a* | *Vrn-D1v*<br>*Vrn-D1v*<br>*Vrn-D1v*<br>*Vrn-D1v* | *Ppd-A1b*<br>*Ppd-A1b*<br>*Ppd-A1b*<br>*Ppd-A1b* | *Ppd-D1a*<br>*Ppd-D1a*<br>*Ppd-D1a*<br>*Ppd-D1a* | *Ppd-B1b*<br>*Ppd-B1b*<br>*Ppd-B1b*<br>*Ppd-B1b* | *Rht-B1a*<br>*Rht-B1a*<br>*Rht-B1a*<br>*Rht-B1a* | *Rht-D1b*<br>*Rht-D1b*<br>*Rht-D1b*<br>*Rht-D1b* |
| 4 | SSW-BA-TD | UK-2 | *Vrn-A1a* | *Vrn-B1a* | *Vrn-D1v* | *Ppd-A1b* | *Ppd-D1b* | *Ppd-B1a* | *Rht-B1a* | *Rht-D1b* |
| 5 | SWS-AB-DT | SP-1 | *Vrn-A1a* | *Vrn-B1v* | *Vrn-D1a* | *Ppd-A1b* | *Ppd-D1a* | *Ppd-B1b* | *Rht-B1b* | *Rht-D1a* |
| 6 | SWW-AA-DT | UK-4 | *Vrn-A1a* | *Vrn-B1v* | *Vrn-D1v* | *Ppd-A1b* | *Ppd-D1a* | *Ppd-B1c* | *Rht-B1b* | *Rht-D1a* |
| 7 | SWW-BA-TD | UK-1 | *Vrn-A1a* | *Vrn-B1v* | *Vrn-D1v* | *Ppd-A1b* | *Ppd-D1b* | *Ppd-B1a* | *Rht-B1a* | *Rht-D1b* |
| 8 | WSS-AA-DT | SP-4<br>CMT-3 | *Vrn-A1v*<br>*Vrn-A1v* | *Vrn-B1a*<br>*Vrn-B1a* | *Vrn-D1a*<br>*Vrn-D1a* | *Ppd-A1b*<br>*Ppd-A1b* | *Ppd-D1a*<br>*Ppd-D1a* | *Ppd-B1a*<br>*Ppd-B1a* | *Rht-B1b*<br>*Rht-B1b* | *Rht-D1a*<br>*Rht-D1a* |
| 9 | WSS-AB-DT | CMT-2<br>QLD-3 | *Vrn-A1v*<br>*Vrn-A1v* | *Vrn-B1a*<br>*Vrn-B1a* | *Vrn-D1a*<br>*Vrn-D1a* | *Ppd-A1b*<br>*Ppd-A1b* | *Ppd-D1a*<br>*Ppd-D1a* | *Ppd-B1b*<br>*Ppd-B1b* | *Rht-B1b*<br>*Rht-B1b* | *Rht-D1a*<br>*Rht-D1a* |
| 10 | WSS-AB-TD | Mace | *Vrn-A1v* | *Vrn-B1a* | *Vrn-D1a* | *Ppd-A1b* | *Ppd-D1a* | *Ppd-B1b* | *Rht-B1a* | *Rht-D1b* |
| 11 | WSS-BA-DT | CMT-1 | *Vrn-A1v* | *Vrn-B1a* | *Vrn-D1a* | *Ppd-A1b* | *Ppd-D1b* | *Ppd-B1a* | *Rht-B1b* | *Rht-D1a* |
| 12 | WSW-AB-DT | QLD-2<br>SP-3 | *Vrn-A1v*<br>*Vrn-A1v* | *Vrn-B1a*<br>*Vrn-B1a* | *Vrn-D1v*<br>*Vrn-D1v* | *Ppd-A1b*<br>*Ppd-A1b* | *Ppd-D1a*<br>*Ppd-D1a* | *Ppd-B1b*<br>*Ppd-B1b* | *Rht-B1b*<br>*Rht-B1b* | *Rht-D1a*<br>*Rht-D1a* |
| 13 | WSW-AB-TD | Wyalkatchem<br>Magenta | *Vrn-A1v*<br>*Vrn-A1v* | *Vrn-B1a*<br>*Vrn-B1a* | *Vrn-D1v*<br>*Vrn-D1v* | *Ppd-A1b*<br>*Ppd-A1b* | *Ppd-D1a*<br>*Ppd-D1a* | *Ppd-B1b*<br>*Ppd-B1b* | *Rht-B1a*<br>*Rht-B1a* | *Rht-D1b*<br>*Rht-D1b* |
| 14 | WSW-BB-DT | UK-3 | *Vrn-A1v* | *Vrn-B1a* | *Vrn-D1v* | *Ppd-A1b* | *Ppd-D1b* | *Ppd-B1b* | *Rht-B1b* | *Rht-D1a* |
| 15 | WWW-AB-DT | QLD-1 | *Vrn-A1v* | *Vrn-B1v* | *Vrn-D1v* | *Ppd-A1b* | *Ppd-D1a* | *Ppd-B1b* | *Rht-B1b* | *Rht-D1* [i] |

[i] First three letters represent status of *Vrn A1, Vrn B1*, and *Vrn D1* loci; next two letters represent *Ppd D1* and *Ppd B1* loci; last two letters represent *RhtB1* and *RhtD1*.

**Table 2.** List of the parents in different gene pools.

| Pool | Descriptions | Genetic Background |
|---|---|---|
| Spanish/French (SP) | Biparental cross | Califa Sur, Rinconda, Farak, Arrturnik, Fidel, Soissons, Recital, Monopol |
| Winter x Spring (UK) | Three-way | Winter: Einstein, WW66, Heperion (from Europe) Spring: Correl, Carinya, VP1081, Sunzell, Sunstate (Australia) |
| Synthetic Hexaploid (VIC) | Multi-parental | Pavon, 30271, TM56, Janz, Annuello, *Aegilops squarrosa* |
| CIMMYT (CMT) | Multi-parental and wide crosses | Kiritati, Waxwing, Onix, Tacupeto, Pastor, CRBD-3, Stork, *T. diccoides* |
| Queensland (QLD) | Recurrent selection | Seri, Batavia, Kukri, Sunstate, Janz, Hartog |

**Table 3.** Location and allelic combination interaction effects on yield and protein content.

| Sl No. | ALLELIC COMBINATION | Number of Lines | YIELD (ton/Ha) | | | PROTEIN (%) | | |
|---|---|---|---|---|---|---|---|---|
| | | | Corrigin | Kojonup | Toodyay | Corrigin | Kojonup | Toodyay |
| 1 | SSSABDT | 1 (CMT-4) | 2.84 | 4.87 | 3.78 | 10.53 | 11.57 | 12.72 |
| 2 | SSWABDT | 2 (QLD-4 and Bonnie Rock) | 2.98 | 4.75 | 3.42 | 10.32 | 12.77 | 11.36 |
| 3 | SSWABTD | 4 (SP-2, VIC-1, VIC-2 and VIC-3) | 3.19 | 5.27 | 4.74 | 10.24 | 11.81 | 10.74 |
| 4 | SSWBATD | 1 (UK-2) | 3.37 | 5.32 | 3.41 | 9.87 | 11.70 | 11.79 |
| 5 | SWSABDT | 1 (SP-1) | 3.00 | 4.60 | 3.73 | 11.57 | 13.27 | 11.42 |
| 6 | SWSAADT | 1 (UK-4) | 3.04 | 5.15 | 3.43 | 10.20 | 10.70 | 11.90 |
| 7 | SWWBATD | 1 (UK-1) | 2.95 | 4.77 | 3.66 | 9.80 | 11.17 | 11.83 |
| 8 | WSSAADT | 2 (SP-4 and CMT-3) | 2.70 | 4.93 | 4 | 10.42 | 11.98 | 11.65 |
| 9 | WSSABDT | 2 (CMT-2 and QLD-3) | 2.90 | 5.19 | 3.89 | 10.53 | 11.92 | 11.60 |
| 10 | WSSABTD | 1 (Mace) | 3.65 | 5.04 | 5.39 | 9.50 | 12.53 | 10.48 |
| 11 | WSSBADT | 1 (CMT-1) | 3.15 | 4.56 | 3.59 | 9.93 | 11.93 | 11.33 |
| 12 | WSWABDT | 2 (QLD-2 and SP-3) | 2.74 | 5.37 | 3.87 | 10.50 | 11.87 | 11.33 |
| 13 | WSWABTD | 2 (Wyalkatchem and Magenta) | 3.20 | 5.15 | 3.35 | 10.26 | 12.08 | 12.25 |
| 14 | WSWBBDT | 1 (UK-3) | 2.88 | 4.62 | 3.03 | 10.37 | 10.90 | 11.26 |
| 15 | WWWABDT | 1 (QLD-1) | 2.33 | 4.66 | 4 | 10.70 | 11.13 | 11.09 |
| | Standard Error of Difference (SED)± | | | 0.2929 | | | 0.6729 | |

## 3. Results

### 3.1. Allelic Distribution at The Vrn1 and Ppd loci

The nineteen advanced lines and four local checks were genotyped to determine the individual allelic combination of phenology and dwarfing genes. The spring allele *Vrn-A1a* was identified in 10 lines with a frequency of 43.48% (Table 1 and Figure S1). Only one line (Vic-3) produced a 522-bp product, indicating the presence of a Langdon-type spring allele *Vrn-A1c* [26]. On the other hand, a 1068-bp fragment was amplified in the remaining 12 lines, indicating the presence of the recessive *Vrn-A1v* allele. The frequency of the dominant *Vrn-B1a* allele was 82.61%, identified in 19 lines and the remaining four lines produced a 1149-bp fragment characteristic of the recessive *Vrn-B1v* allele. Amplification of a 1671-bp fragment characteristic of the *Vrn-D1a* allele was produced in eight lines, while a 997-bp product characteristic of recessive *Vrn-D1v* was generated in the remaining 15 lines. All the lines contained the recessive allele of *Ppd-A1b*. Only one line (UK-4) produced a 994-bp fragment characteristic of the four-copy *Ppd-B1* of Chinese Spring. On the other hand, a 223-bp fragment characteristic of the three-copy allele *Ppd-B1a* of Sonora 64 was produced in five lines. Nineteen lines produced a 218-bp fragment of the photoperiod-insensitive *Ppd-D1a* allele and the remaining four lines produced a 414-bp product of the photoperiod-sensitive *Ppd-D1b* allele (Table 1). Fourteen lines had the *Rht-B1b* and the remaining nine lines had the *Rht-D1b* allele with frequency distributions of 60.87% and 39.13% (Table 1).

### 3.2. Environmental Effects on Yield and Protein Content

All 23 lines were grown in three different locations in Western Australia. The mean grain yield across the locations ranged from 2.99 to 4.95 t/ha and protein content ranged from 10.32% to 11.82%. Grain yield at the Kojonup trial site was 4.95 t/ha and significantly ($p < 0.05$) higher than the other two trial sites of Toodyay and Corrigin, which had similar yields (Figure 1 and Table S2). Protein content at

Kojonup and Toodyay was statistically similar, but for Corrigin, at 10.32%, it was significantly lower than the other sites (Figure 1 and Table S3).

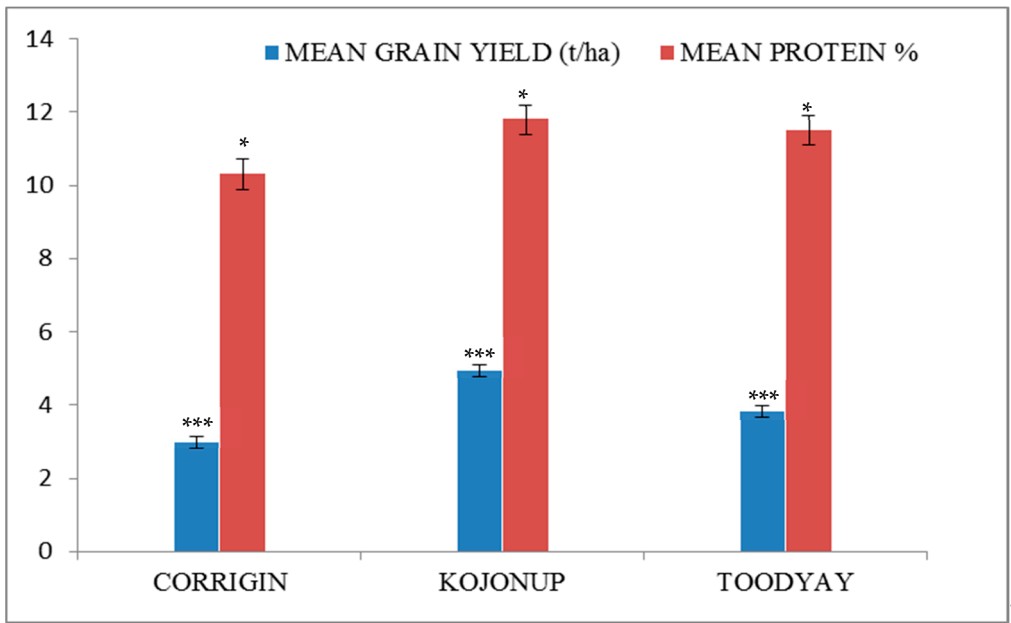

**Figure 1.** Location means of grain yield, and protein content for the 23 lines grown at the three trial sites. The blue bar represents the mean yield (t/ha) of each location and the maroon bar represents the mean protein % of each location. For yield, Kojonup is significantly higher than Toodyay. Corrigin is significantly lower than Toodyay and Kojonup for both yield and protein %, respectively. [* = $p < 0.05$%, and *** = $p < 0.001$ level of significance]

*3.3. Environmental and Allelic Combination Effects on Yield and Protein Content*

Considering the dominant and recessive alleles at the *Vrn-1*, *Ppd-1*, *Rht-B*, and *Rht-D* loci, the 23 lines investigated were grouped into 15 classes, where the commercial variety Bonnie Rock shared a similar allelic combination with only one line; both Wyalkatchem and Magenta shared a similar allelic combination with another line, while Mace and another nine lines showed unique allelic combinations (Table 3).

The mean grain yield (GY) at Kojonup ranged from 4.56 to 5.37 t/ha for different allelic variants, followed by Toodyay, ranging from 3.03 to 5.39 t/ha, and Corrigin from 2.33 to 3.65 t/ha (Table 3). The mean GY significantly varied among the different allelic variants for each trial site and also among the trial sites for each allelic group (Table S2). Control WSSABTD (Mace) was the highest yielder in Corrigin and Toodyay, with a mean GY of 3.65 and 5.39 t/ha, respectively. The mean GY of Mace at Kojonup (5.04 t/ha) did not differ significantly from the highest mean GY (5.37 t/ha) by the allelic variant WSWABDT, although the same allelic variant gave low GY in Corrigin at 2.74 t/ha (Table 3). The mean GY for the allelic combination SSWABTD across the three locations, Corrigin, Kojonup, and Toodyay, was 3.19, 5.27, and 4.74 t/ha, respectively, where the difference was insignificant compared to the highest yielder at the respective location. In contrast, the allelic combination WWWABDT gave significantly lower GY compared to the highest yielder in all the three locations—2.33, 4.66, and 4 t/ha for Corrigin, Kojonup, and Toodyay, respectively.

Protein content across the trial sites ranged from 9.5 to 11.57% in Corrigin, 10.70 to 13.27% in Kojonup, and 10.48 to 12.72% in Toodyay (Table 3). A significant variation in protein content was observed among the allelic groups within and across trial sites (Table S3). The allelic combination SWS-SW-DT had the highest protein content in Corrigin and Kojonup, 11.57 and 13.27% respectively, and also close to the highest in Toodyay, which has the allelic combination of SSSABDT and protein at 12.72%.

For ASV value, allelic combination SSSABDT is the most stable for yield across the environments, followed by SWWBATD and WSSABDT. On the other hand, considering both bi and $S^2$di value, combinations SSSABDT and WSWABTD are the most adaptable and stable genotypes (Table 4). Meanwhile, SSWABTD and SSWBATD are considered as both high yielding and widely adaptable. A regression coefficient value greater and smaller than one denotes the higher response to high yielding environments and better resistance to environmental changes. According to this, WSWABTD is adapted to high yielding environments (5.37 t/ha in Kojonup) and WSSABTD is suitable for a wide range of environments (Table 4).

The stability parameters for the protein content of the allelic combinations are presented in Table 4. According to ASV measurement, WSSAADT and WSSABDT are the most stable allelic combination for protein content. Based on both regression coefficient one (bi) and squared deviation from regression ($S^2$di), WSSAADT is also the most stable and widely adaptable allelic combination. On the other hand, SSWBATD, WSWABTD, and SSWABDT are better adapted to favourable environments, and in contrast, SWSABDT and WSWABDT are more suited to stressed environments.

A biplot was constructed to obtain the allelic combination by location interaction effects for both yield and protein content (Figure 2A,B). The GY positive and negative values in both axes indicated that some allelic combinations had a positive interaction with one or two locations and a negative interaction with others. From the biplot, it was observed that the best three allelic combinations for each site were: WSWABDT, SSWBATD, and SSWABTD for Kojonup; WSSABTD, SSWBATD, and WSWABTD for Corrigin, and WSSABTD, SSWABTD, and WSSAADT for Toodyay (Figure 2A). An allelic combination and location interaction was also observed for protein content. The biplot showed that the best allelic combinations for each site were SSSABDT, WSWABTD, and SWWBATD for Toodyay, SWSABDT, WWWABDT, and SSSABDT for Corrigin, and SWSABDT, SSWABDT, and WSSABTD for Kojonup (Figure 2B).

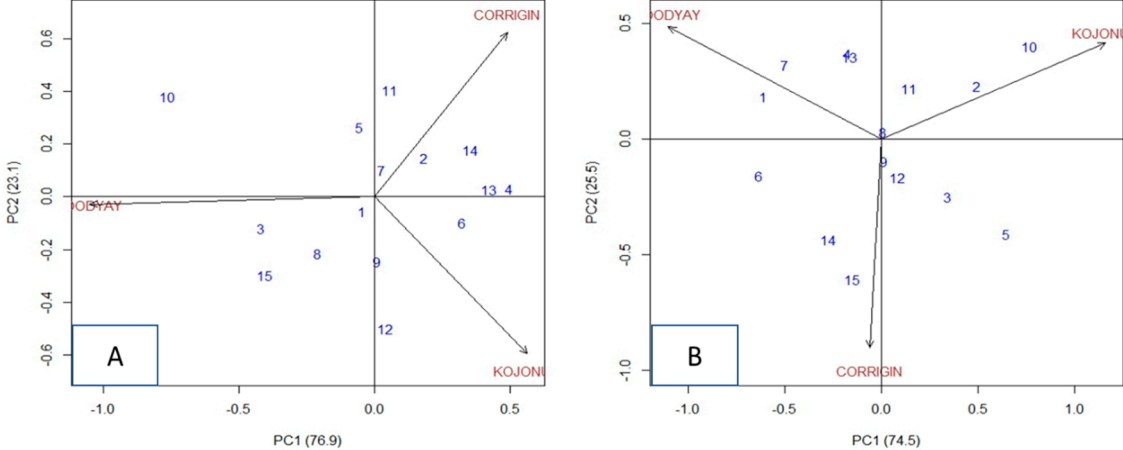

**Figure 2.** Additive main effects and multiplicative interactions (AMMI) biplot for allelic combination and environmental interaction on (**A**) yield and (**B**) protein content. Genotypes positioned near the origin of the biplot represent stability across the environments and close to an environment represent best performance in that corresponding environment.

**Table 4.** Stability parameters and environmental effects on the allelic combinations for yield and protein content.

| Sl No | Allelic Combination | Yield (ton/Ha) | | | | GEI | | | Protein | | | | GEI | | |
|---|---|---|---|---|---|---|---|---|---|---|---|---|---|---|---|
| | | ASV (Rank) | $S^2di$ | bi | Mean | CORRIGIN | KOJONUP | TOODYAY | ASV (Rank) | $S^2di$ | bi | Means | CORRIGIN | KOJONUP | TOODYAY |
| 1 | SSSABDT | 0.101 (1) | 0.080 | 1.033 | 3.83 | −0.062 | 0.011 | 0.050 | 1.054 (12) | 0.773 | 1.038 | 11.611 | −0.150 | −0.622 | 0.772 |
| 2 | SSWABDT | 0.362 (5) | 0.022 | 0.922 | 3.72 | 0.190 | 0.006 | −0.196 | 0.877 (10) | 0.303 | 1.371 | 11.506 | −0.261 | 0.683 | −0.423 |
| 3 | SSWABTD | 0.773 (12) | 0.219 | 1.031 | 4.40 | −0.288 | −0.157 | 0.445 | 0.642 (8) | 0.134 | 0.828 | 10.944 | 0.228 | 0.289 | −0.517 |
| 4 | SSWBATD | 0.905 (14) | 0.324 | 1.040 | 4.03 | 0.264 | 0.260 | −0.524 | 0.476 (6) | 0.100 | 1.344 | 11.156 | −0.361 | −0.033 | 0.394 |
| 5 | SWSABDT | 0.285 (4) | 0.083 | 0.812 | 3.78 | 0.155 | −0.207 | 0.053 | 1.182 (14) | 1.021 | 0.736 | 12.122 | 0.373 | 0.567 | −0.940 |
| 6 | SWSAADT | 0.598 (9) | 0.078 | 1.105 | 3.87 | 0.092 | 0.246 | −0.338 | 1.094 (13) | 0.703 | 0.671 | 10.933 | 0.195 | −0.811 | 0.616 |
| 7 | SWWBATD | 0.112 (2) | 0.083 | 0.934 | 3.79 | 0.083 | −0.052 | −0.031 | 0.913 (11) | 0.514 | 1.211 | 11.022 | −0.294 | −0.433 | 0.727 |
| 8 | WSSAADT | 0.437 (7) | 0.002 | 1.121 | 3.88 | −0.250 | 0.024 | 0.227 | 0.037 (1) | 0.230 | 1.037 | 11.378 | −0.033 | 0.028 | 0.005 |
| 9 | WSSABDT | 0.247 (3) | 0.084 | 1.169 | 3.99 | −0.165 | 0.168 | −0.003 | 0.099 (2) | 0.228 | 0.903 | 11.367 | 0.095 | −0.028 | −0.067 |
| 10 | WSSABTD | 1.449 (15) | 0.790 | 0.648 | 4.69 | −0.114 | −0.683 | 0.796 | 1.368 (15) | 1.019 | 1.622 | 10.878 | −0.450 | 1.078 | −0.628 |
| 11 | WSSBADT | 0.418 (6) | 0.070 | 0.730 | 3.77 | 0.308 | −0.236 | −0.072 | 0.327 (4) | 0.199 | 1.266 | 11.089 | −0.227 | 0.267 | −0.040 |
| 12 | WSWABDT | 0.505 (8) | 0.085 | 1.347 | 3.99 | −0.328 | 0.352 | −0.024 | 0.221 (3) | 0.200 | 0.847 | 11.267 | 0.162 | 0.022 | −0.184 |
| 13 | WSWABTD | 0.766 (11) | 0.209 | 1.031 | 3.90 | 0.226 | 0.217 | −0.444 | 0.454 (5) | 0.113 | 1.330 | 11.530 | −0.346 | −0.030 | 0.375 |
| 14 | WSWBBDT | 0.668 (10) | 0.140 | 0.922 | 3.51 | 0.295 | 0.083 | −0.378 | 0.641 (7) | 0.115 | 0.473 | 10.844 | 0.450 | −0.522 | 0.072 |
| 15 | WWWABDT | 0.798 (13) | 0.226 | 1.154 | 3.66 | −0.405 | −0.032 | 0.437 | 0.659 (9) | 0.220 | 0.323 | 11.011 | 0.617 | −0.455 | −0.162 |
| | Mean | | | | 3.92 | | | | | | | 11.24 | | | |

### 3.4. Allelic Combination Effects on Agronomic Traits

Data for days to heading, plant height, and other agronomic traits were recorded from the Toodyay trial site and analysed to obtain any significant differences among the allelic groups for the traits of interest. Days to heading (DH) ranged from 93 to 110 days, with most allelic combinations having a DH value of 102 days (Figure 3A). Allelic combination SSWABTD took 93 days to heading, followed by WSWABDT (96 days) and SSWABDT and WSSABTD (97 days). On the other hand, allelic combination WSWBBDT took 110 days to heading, followed by SWWBATD (107 days). The presence of double spring alleles of *Vrn-A1a + Vrn-B1a* and *Vrn-B1a + Vrn-D1a* showed additive effects and advanced the heading time by 5 days and 3.3 days, respectively, whereas the presence of double spring alleles of *Vrn-A1a + Vrn-D1a* showed epistatic effects delayed the heading time up to 3 days. The highest reduction in heading observed due to the presence of the photoperiod-insensitive allele of *Ppd-D1a*, reducing the heading time by 12.67 and 7.5 days compared to the presence of both sensitive alleles and the insensitive *Ppd-B1a* allele, which is consistent with earlier reports [34,35]. The presence of both photoperiod-insensitive alleles showed additive effects and reduced the heading time by 4 days (Figure 3A), also reported in the previous studies [35]. Plant height of the allelic variants ranged from 70 to 100 cm, with more than half of the allelic groups having a height of 83–85 cm (Figure 3B). Two allelic groups SSWBATD and WSWBBDT were significantly shorter, being 70 and 75 cm, respectively. In contrast, the four allelic groups WSSBADT, WSWWDT, SSSABDT, and WSSAADT were significantly taller than the average, being greater than 95 cm.

Spike length of the allelic groups ranged from 7.6 to 10.9 cm, whereby four allelic groups, namely SSSABDT, WSSABDT, SWSABDT, and WSWABDT, had values above 10.5 cm and two allelic groups, SWSAADT and WSWBBDT, were shorter, being 8.7 and 7.5 cm, respectively (Figure 3C). Seed number per spike ranged from 43 to 76 among the allelic groups, whereby allelic groups SWSABDT and SSWABTD had the highest number of seeds per spike and allelic groups SSWABDT and WSWBBDT had the lowest number of seeds per spike (Figure 3D).

Seed number per spike and spike length ratio were calculated to estimate spikelet fertility. It was observed that seeds per unit spike length ranged from 4.7 to 8.2, whereby SSWABTD had the highest value, followed by SWSABDT, and SSWABDT had the lowest value followed by WSSBADT (Figure 3E). Thousand-kernel weight (TKW) for the allelic groups ranged from 31.3 to 41.8 g, whereby most groups had values within a range of 35 to 39 g (Figure 3F). The aspect ratio of the allelic group ranged from 1.8 to 2.1, and roundness ranged from 0.59 to 0.97 (Figure 3G,H).

Overall, the allelic combination SSWABTD showed better performance for yield and stability, mostly contributed by early maturity, short stature, high grain number per spike, and spikelet fertility as well as average spike length and Thousand Kernel Weight (TKW). These lines showed the ability to produce good yields at the lowest moisture level site (Corrigin) but showed the ability to respond well to the more favourable environment at Kojonup.

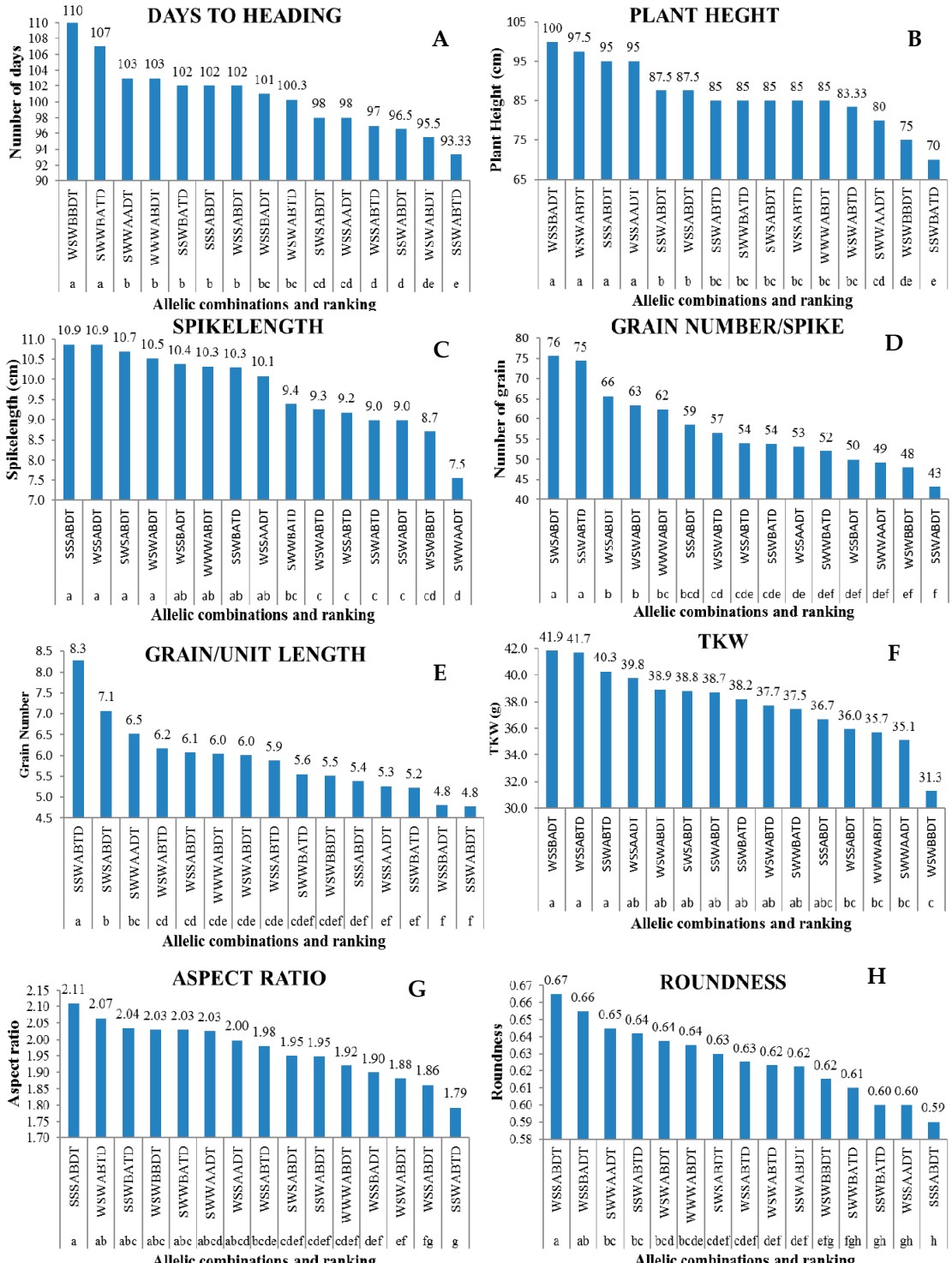

**Figure 3.** Comparison of the allelic combination effects on (**A**) days to heading; (**B**) plant height; (**C**) spike length; (**D**) grain number per spike; (**E**) spikelet fertility (grain per unit spike length); (**F**) thousand-kernel weight; (**G**) aspect ratio; and (**H**) roundness. Letters at the bottom of each allelic combination indicate level of significant.

## 4. Discussion

### 4.1. Allelic Diversity in the Advanced Lines Used

This study focused on the effects of allelic interactions at the vernalization (*Vrn-1*), photoperiod (*Ppd-1*), and reduced height (*Rht-1* and *Rht-2*) loci for adaptation to three different wheat growing environments of Western Australia. A set of 19 advanced breeding lines were used, together with four leading commercial check cultivars. The genotypes used formed clusters of diverse genetic backgrounds, each with their own set of allelic combinations. Having diverse sets of genotypes made it possible to study the genetic effects of various allele combinations within a specific genetic background, as well as the interactions of the two factors. Each genotype was characterised for the allelic variant combination at the four loci. For the *Vrn-A1* locus, the temperature-insensitive Vrn-A1a and the sensitive *Vrn-A1v* alleles were almost equally distributed among all lines. The one exception (Vic 3) was a synthetic wheat variety derived from a cross onto 'Annuello' (which has a Victorian germplasm genetic background) and this line carried the weaker spring allele *Vrn-A1c*, characteristic of the Langdon durum parent used to make the synthetic (Table 1). The synthetics were made by crossing durum wheat varieties and the wild wheat species *Aegilops squarrosa*. The latter was found to confer early maturity in the combinations selected for the breeding program, and when crossed onto 'Annuello', the three early-maturing lines used in this study were obtained. The weaker spring allele (*Vrn-A1c*) delayed the heading by two days compared to other members of the same allelic group, and this affected the yield and stability (Table S4). Spanish germplasm has also been widely used in the Edstar wheat breeding program (Table 2) by one of the authors, and a number of crosses to these lines have provided varieties of early maturity and good dryland adaptability (personal observation: Ian Edwards) [36]. Although *Vrn-A1a* has a stronger effect on vernalization requirement than *Vrn-B1a* [37,38], the reduction in days to heading in lines containing *Vrn-B1a* likely contributed to this, being the most frequently observed spring allele among the varieties, regardless of the genetic background, thereby indicating its broad adaptive value. Earlier maturity has been found to have a positive impact on grain yield in water-limited environments, and previous work has suggested that this may be a key reason for the higher frequency of this gene among lines that perform well under moisture stress [39,40].

None of the lines in this study were found to have a photoperiod-insensitive allele at the *Ppd-A1* locus, although it has been previously reported that earlier flowering is associated with *Ppd-A1* when compared with those having the other photoperiod-insensitive allele at the *Ppd-B1* locus [34,35]. This would suggest that further gains might be made in the future through incorporation of this gene into the breeding program and evaluating its effect on yield and adaptive value. For the *Ppd-D1* locus, *Ppd-D1a* was the most frequently found photoperiod-insensitive allele in all lines, and this allele contributed to the earliest flowering [35]. The notable exceptions were lines derived from the UK winter x spring genetic background and one line from the CIMMYT wheat program (CMT-1). In this study, great allelic variation existed in different genetic backgrounds except for the Victorian background, which traces to a common synthetic hexaploid source where all the lines had the same dominant and recessive allelic combinations for all loci—the one exception having a different spring allele at the *Vrn-A1* locus (Table 1).

### 4.2. Allelic Combination Effects on Agronomic Traits

The allelic combination effect of phenology and reduced height genes was investigated for one site (Toodyay) on heading and a few other agronomic traits. Final yield is the cumulative result of several successful events during the plant's developmental phases. Heading date, which is one of the most important considerations for water-limited environments, varied significantly among the allelic combinations. In this study, the *Vrn1* gene has been found to have a significant effect on heading date, which has not been reported previously. Additive gene effects were observed of reducing heading time for double spring alleles of *Vrn-A1a* + *Vrn-B1a* and *Vrn-D1a* + *Vrn-B1a* but epistatic interactions

were observed for double spring alleles of *Vrn-A1a* + *Vrn-D1a* and triple spring alleles, which has been also reported earlier also [41]. It was observed that the two spring alleles in the *Vrn1* loci in combination with an insensitive allele in *Ppd-D1* were the earliest in terms of heading. This result is in accordance with previous individual studies, where genotypes with dominant alleles in *Vrn-A1* and *Ppd-D1* loci resulted in an early flowering [12,24,41]. Considering all the loci fixed for *Vrn1* and *Rht*, earlier heading occurred in the presence of the *Ppd-D1a* allele, which indicates that *Ppd-D1a* has stronger effects in reducing heading time compared to *Ppd-B1a*. This current result is in agreement with some previous studies [35,42]. A number of previous studies reported that *Vrn1* genes have a limited role on reproductive development after the floral primordia initiation stage [43,44] and thus, later stages are mainly controlled by the *Ppd* genes [9,45]. A recent study by Grogan et al. [24] also suggested that flowering is more strongly influenced by photoperiod than vernalization genes. In the current study, up to 7 days differences in heading have been recorded due to the variation in *Vrn1* loci under the same *Ppd* and *Rht* allele background. On the other hand, up to 9 days difference was observed due to the variation in *Ppd* loci under the same *Vrn1* and *Rht* allele background. The *Rht* genes were also found to have a significant effect on heading. Grogan et al. [24] reported the effect of *Rht-B1b* and *Rht-D1b* for 2.4 and 2.9 days earlier heading, respectively, across environments. The current results showed up to 5 days differences due to variation in the *Rht* loci. Wilhelm et al. [46] also suggested the association of *Rht* with *Ppd* in determining the heading time and plant height.

Plant height is also an important trait for drought adaptability, as the stem supplies stored carbohydrate assimilates to grains during drought [47]. All the lines in this study were semi-dwarfs with an intermediate plant height of 70 to 100 cm (Figure 3B). This result indicated that selection of breeding lines for WA environments was in accordance with previous findings that a plant height of 70–100 cm maximised yield across environments [48]. Among the 15 allelic combinations, only three pairs provided the opportunity to compare the effects of *Rht* alleles in interaction with vernalization and photoperiod alleles (SSWABDT vs. SSWABTD, WSSABDT vs. WSSABTD, and WSWABDT vs. WSWABTD). In all cases, the *Rht-D1b* genotypes produced shorter plants and a better or similar response to the yield and other agronomic traits. A comparison between the two pairs of allelic groups SSWABTD vs. SSWABDT and WSSABDT vs. WSSABTD, which differed only for the reduced height allele, revealed the superiority of *Rht-D1b* over *Rht-B1b* for early flowering, plant height, seed number per spike, and spikelet fertility, which was also reflected in the plot yield at Toodyay and two other sites. This result is in agreement with Eagles et al. [40] who reported that the *Rht-B1a/Rht-D1b* combination was advantageous in lower rainfall areas where drought stress and high temperatures coincide during flowering and grain-filling periods. However, by comparing WSWABTD vs. WSWABDT, *Rht-B1b* showed early heading and better agronomic performance. It was also observed that lines having similar alleles for *Rht-B1* and *Rht-D1* but varying alleles in *Vrn1* and *Ppd* loci had significant differences in plant height. This study identified up to 7 cm difference due to the variation either in *Vrn* or *Ppd* loci. These results revealed the interactions of these three developmental pathways in determining plant height and thus, yield, demonstrating that the *Rht* gene interacts significantly with other phenology genes in affecting a range of agronomic traits.

Grain number, determined by spike length and grain number per spike, is the most important yield determining factor [49]. Different allelic groups showed significant variations in spike length and grain number per spike. This study revealed that the allelic groups with larger spikes did not always produce the highest number of grains per spike, which is the reason why spikelet fertility as a selection criterion has been emphasised in the breeding program to increase grain yield [50,51]. In this study, ratio of grain number to spike length has been used as an indicator of spikelet fertility, although other studies have used the ratio between grain number and spike chaff dry [52]. The current results demonstrated that lines with two spring alleles in *Vrn1* loci with an insensitive allele in *Ppd-D1a* produced the higher number of grains per spike and greater ratio of grain number to spike length. This has revealed that most of the early flowering lines performed better than the late flowering allelic combination group. This was probably as a consequence of floret abortion and/or sterile grain

due to the effect of drought stress on the late flowering lines. A similar trend was observed for the thousand-kernel weight, where the allelic group with two spring alleles at the *Vrn1* loci had a higher TKW than the allelic group with two or three winter alleles.

*4.3. Environment and Allelic Combination Effects on Yield and Protein Content*

A significant variation in yield performance and protein content was observed in most of the allelic groups across locations, whereby the Kojonup results were the highest and Corrigin the lowest among the three environments (Figure 1). There was not much variation in terms of total rainfall during the cropping season among the three locations. Kojonup and Corrigin received almost the same amount of rainfall, while Toodyay received only 15 mm more rainfall than the two other sites. Regarding the monthly average temperature, Toodyay had the highest monthly average temperature, closely followed by Corrigin, and Kojonup was almost 1.5 °C less than the other two locations (Table S1). The higher temperature in Corrigin and Toodyay led to higher pan evaporation and evapotranspiration and aggravated the drought effects in these two locations. Thus, the yield benefits at the Kojonup site could be explained by these temperature differences. Again, as Toodyay received more rainfall than Corrigin but had a similar range in temperature, the former produced a better yield than the latter.

Based on the stability parameter analysis, the performance of the same allelic group for yield and protein also varied across environments, which resulted from varying environmental stimuli like temperature, soil moisture, and day length (Table 2 and Figure 2). These GxE interactions demonstrate the importance of selecting and identifying stable high yielding genotypes [53]. In the current study, the top three ASV ranked allelic combinations showed the least GxE interaction across the environments, indicating that the ASV parameter is a good indicator for selecting stable genotypes (Table 3). Meanwhile, according to Eberhart and Russell [32], genotypes having a regression coefficient equal to unity (bi = 1) coupled with small deviation from regression ($S^2$di = 0) and higher than mean yield should be considered as stable. Therefore, simultaneous selection of both the yield and stability parameters is the logical way to overcome GxE effects and obtain adaptable genotypes. Lines having all the spring alleles in the *Vrn-1* loci along with *Ppd-D1a* ranked top in ASV and also showed regression coefficients close to one (1.033) and a small deviation from regression ($S^2$di = 08), indicating low GxE effects. However, their yield was below average (Table 3). Lines with two spring alleles in *Vrn-1* loci and one insensitive allele in *Ppd* loci can be considered as better allelic combinations based on both yield and stability parameters. On the other hand, WSSABTD showed high grain yield and GxE interaction; therefore, this can be recommended for a high yielding environment which is also supported by bi and $S^2$di values.

In this study, it was found that the interaction of *Rht-D1b* with photoperiod-insensitive *Ppd-D1a* and at least two spring-type alleles of *Vrn1* loci performed the best for grain yield across the environments (Table 3). It is interesting to note that the lines with two dominant alleles at the homoeologous *Vrn1* loci and one dominant allele at the *Ppd* loci or vice versa, in combination with the reduced height allele *Rht-D1b*, had considerably more stable yields across environments (Table 3). These observations are supported by the previous study of Eagles et al. [40] that concluded that genotypes with *Rht-B1a/Rht-D1b* are advantageous for yield in most environments. In addition to this, *Ppd-D1a* and *Vrn-A1a* have been reported to induce early flowering [12,41], which might result in better yield due to greater incident radiation during grain-filling and avoidance of terminal drought. In contrast, lines with two or three winter alleles at the *Vrn1* loci (WWWABDT, WSWBBDT, and SWWBATD) in combination with the same photoperiod and reduced-height alleles produced lower yields across environments. This might be the consequence of later heading in those lines due to longer vernalization period requirements and were likely affected by terminal drought stress. The higher yielding lines with winter alleles at the *Vrn-A1* and *Vrn-D1* loci combining *Ppd-A1a* and *Rht-B1b* (WSWABDT) at Kojonup could be the consequence of the more favourable environmental conditions.

Protein content also varied significantly among trial sites as well as among the different allelic combination groups (Table 3). The higher protein content in the Kojonup trial could be explained by

soil fertility and more suitable environmental conditions during the growing season. Protein content for different allelic groups varied significantly, indicating that the phenology and reduced height genes can significantly influence yield and the protein content of a cultivar. In general, allele groups with high protein content showed lower grain yield. Interestingly, some high-yielding lines also had high protein content. Previous studies indicated that GxE contributes to the protein content more than genotypes [54]. However, in the current study, the stability analysis revealed a few stable and widely adaptable allelic combinations for protein content and grain yield, including WSSABDT and WSSAADT, a clear indication that a proper combination of the phenology and the reduced height genes can minimise the GxE effects on protein content. Since a stable protein content along with high grain yield is the prime breeding goal, the genetic effects of the phenology and reduced height genes on protein content deserve more attention.

The advanced lines studied here were developed from diverse genetic backgrounds but had gone through the standard selection processes based on yield, rust resistance, maturity, height, grain quality, and overall agronomic performance for the Australian agro-climatic conditions. It was assumed that the lines contained common genes for most of the standard traits as screened under Australian conditions but varied in maturity. However, the allelic combination effects of the phenology genes on yield and adaptability found in this study included several advanced lines derived from diverse genetic backgrounds, and one cannot preclude the possible presence of some accompanying background effects. To check whether or not the predicted allelic combinations had some background effects, an analysis of the individual lines for stability parameters has been presented in Supplementary Table S4. Analysis of the individual lines for yield and stability parameters also revealed similar results in accordance with their allelic combinations, i.e., lines having the similar allelic combinations produced nearly the same yield. The only exception was SP-3 and QLD-2, which had the same allelic combination but significantly different yield. SP-3, a line derived from a Spanish-French germplasm, produced an outstanding yield with only one spring *Vrn-1* allele, despite the study showing that in most lines, two spring *Vrn-1* alleles performed better. This might be due to some minor gene effects.

## 5. Conclusions

This study characterised a set of advanced lines developed from diverse genetic backgrounds, which enabled detailed analyses of individual allele effects, the comparison of allelic combinations and background genetic interactions, and allelic combination by environment interactions. The focus was on important loci known to affect growth and development via vernalization, photoperiod, and reduced-height pathways. It also investigated the interaction effects of these three pathways on yield and yield components of wheat. Some novel genetic effects have been identified (e.g., the reduced height genes also affect heading time). This study clearly demonstrated the effect of *Vrn1* loci on heading in addition to their previously recognised role in the initiation of the reproductive stage. It also identified the pleiotropic effect of *Vrn1* and *Ppd* loci on plant height aside from the effect on *Rht* genes. The study also indicated that allelic variants at those loci interacted in complex ways to determine the yield and protein content of wheat, and the effects of allele combinations can also be influenced by varying environmental conditions. Finally, this study identified favourable allelic combinations of these developmental genes for stable grain yield and protein content across water-limited environments. While many of the lines out yielded the checks—Wyalkatchem, Magenta, and Bonnie Rock—none of these lines exceeded mean yield of Mace. However, the allelic combination group SSWABTD was more stable than Mace but did not differ significantly for yield, and this increased stability is a key trait required to develop robust varieties for drier climates. Therefore, this study clearly demonstrated the value of utilizing allelic variants of key phenology genes that provide top-end yield with greater stability across contrasting environmental conditions. Climatic change by its very nature will mean greater environmental fluctuations from year to year, and yield stability will assume an even greater importance in assisting breeders to develop varieties that maximise productivity in water-limited environments.

**Supplementary Materials:** The following are available online at http://www.mdpi.com/2077-0472/10/10/470/s1, Figure S1: Frequency distribution for the identified alleles at different locus, Table S1: Environmental condition of the three trial sites during 2014 along with sowing and harvesting time, Table S2: REML variance components analysis for yield, Table S3: REML variance components analysis for protein content, Table S4: Stability parameters and environmental effects on the advanced lines for yield.

**Author Contributions:** Conceptualization, I.E. and M.A.N.N.U.D.; methodology, M.A.N.N.U.D; software, K.S. and M.A.N.N.U.D.; validation, K.S., M.A.N.N.U.D., I.E., S.I., and W.M.; formal analysis, K.S. and M.A.N.N.U.D.; investigation, M.A.N.N.U.D.; resources, I.E., S.I. and W.M.; data curation, M.A.N.N.U.D.; writing—original draft preparation, M.A.N.N.U.D.; writing—review and editing, S.I., I.E., G.O.H.; K.S. and W.M.; visualization, K.S. and M.A.N.N.U.D.; supervision, I.E., G.H. and W.M.; project administration, M.A.N.N.U.D. and I.E.; funding acquisition, I.E. and W.M. All authors have read and agreed to the published version of the manuscript.

**Funding:** This study was partially funded by the standard operating budget of Edstar Genetics Pty. Ltd. of whom Ian Edwards is the principal.

**Conflicts of Interest:** The authors declare no conflict of interest. Ian Edwards solely owns the company Edstar Genetics Pty. Ltd. and as a one of the supervisors of PhD research project decided to carry out this experiment with partial funding and declares no conflicts of interest.

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
