# Peer review of "Phenology and Dwarfing Gene Interaction Effects on the Adaptation of Selected Wheat (Triticum aestivum L.) Advanced Lines across Diverse Water-Limited Environments of Western Australia"

_agriculture, doi:10.3390/agriculture10100470_

Round 1
Reviewer 1 Report
I congratulate the authors on this body of work investigating allelic effects and gene-environment interactions in wheat. The authors identify some novel genetic effects for reduced height and heading time and overall present some useful reference data for breeders.
Comments:
Abstract: The abstract and title of the paper do not indicate which crop is being studied in the manuscript. Please amend.
Introduction: Very nicely supported rationale for the study.
Results section: The reporting on alleles is too dense with information that belongs in the methods. The primer pair combinations used to identify alleles is not required in the results section. Consider significantly reducing the text here. It is not necessary to discuss products that were or were not amplified in the results. You can reduce this to simply report the present alleles in each line. I actually think that the section presented in the discussion (4.1) does a better job of reporting these results. Consider moving parts of the discussion section to the results and removing some of the overly technical/methodological text in the results section.
-Please indicate with asterisks or another mark which bars are significant in Figure 1.
-A Venn diagram would be an easy way to demonstrate which allelic combinations were significant for each of the analyses presented. E.g. which were significantly associated with higher grain yields across sites, which were associated with protein content, which combinations were predicted in the stability analysis. Please consider a visual representation to aid the readers in interpreting the strengths of the different allelic combinations.
-Could the authors please comment on why the agronomic traits were only studied at one site. It seems that a multi-site analysis would have enabled an analysis of gene x environment effects on these traits which may have varied by site and genetic background just as grain yield varied.
-It would be useful to have a summary/take-home message at the end of the agronomic traits results section to summarise which allelic combinations most contributed across the measured traits.
Discussion: As I commented above, parts of the first section of the discussion (4.1) could be moved to the results section.
In lines 477-478 of the discussion the authors state that the allelic combinations did not significantly boost grain yield over mean yield of Mace. Yet in the next sentence state that the findings show the value of utilising allelic combinations to obtain higher yield. This conclusion does not appear to be supported by the findings. Could the authors please elaborate on this to justify this conclusion.
Thank you.
Reviewer 2 Report
The current manuscript needs to improve a lot. Presently manuscript is bulgy
ANOVA, frequency distribution, and genetic analysis missing
Please refer attached pdf form more comments
Best

Round 2
Reviewer 2 Report
Most of the suggested corrections included
Thank you